# Is Model Ensemble Necessary? Model-based RL via a Single Model with Lipschitz Regularized Value Function

**Ruijie Zheng**[1, §]     **Xiyao Wang**[1, §]     **Huazhe Xu**[2, 3]     **Furong Huang**[1]

[1] University of Maryland, College Park   `{rzheng12, xywang, furongh}@umd.edu`
[2] Tsinghua University   `huazhe_xu@mail.tsinghua.edu.cn`
[3] Shanghai Qi Zhi Institute

## Abstract

Probabilistic dynamics model ensemble is widely used in existing model-based reinforcement learning methods as it outperforms a single dynamics model in both asymptotic performance and sample efficiency. In this paper, we provide both practical and theoretical insights on the empirical success of the probabilistic dynamics model ensemble through the lens of Lipschitz continuity. We find that, for a value function, the stronger the Lipschitz condition is, the smaller the gap between the true dynamics- and learned dynamics-induced Bellman operators is, thus enabling the converged value function to be closer to the optimal value function. Hence, we hypothesize that the key functionality of the probabilistic dynamics model ensemble is to regularize the Lipschitz condition of the value function using generated samples. To test this hypothesis, we devise two practical robust training mechanisms through computing the adversarial noise and regularizing the value network's spectral norm to directly regularize the Lipschitz condition of the value functions. Empirical results show that combined with our mechanisms, model-based RL algorithms with a single dynamics model outperform those with an ensemble of probabilistic dynamics models. These findings not only support the theoretical insight, but also provide a practical solution for developing computationally efficient model-based RL algorithms.

## 1 Introduction

Model-based reinforcement learning (*MBRL*) improves the sample efficiency of an agent by learning a model of the underlying dynamics in a real environment. One of the most fundamental questions in this area is how to learn a model to generate good samples so that it maximally boosts the sample efficiency of policy learning. To address this question, various model architectures are proposed such as Bayesian nonparametric models (Kocijan et al., 2004; Nguyen-Tuong et al., 2008; Kamthe & Deisenroth, 2018), inverse dynamics model (Pathak et al., 2017; Liu et al., 2022), multi-step model (Asadi et al., 2019), and hypernetwork (Huang et al., 2021).

Among these approaches, the most popular and common approach is to use an ensemble of probabilistic dynamics models (Buckman et al., 2018; Janner et al., 2019; Lai et al., 2020; Clavera et al., 2020; Froehlich et al., 2022; Li et al., 2022). It is first proposed by Chua et al. (2018) to capture both the aleatoric uncertainty of the environment and the epistemic uncertainty of the data. In practice, MBRL methods with an ensemble of probabilistic dynamics models can often achieve higher sample efficiency and asymptotic performance than using only a single dynamics model.

However, while the uncertainty-aware perspective seems reasonable, previous works only directly apply probabilistic dynamics model ensemble in their methods without an in-depth exploration of why this structure works. There still lacks enough theoretical evidence to explain the superiority of probabilistic neural network ensemble. In addition, extra computation time and resources are needed to train an ensemble of neural networks.

---

[§]Equal contribution

In this paper, we provide a new perspective on why training a probabilistic dynamics model ensemble can significantly improve the performance of the model-based algorithm. We find that the ensemble model-generated state transition samples that are used for training the policies and the critics are much more "diverse" than samples generated from a single deterministic model. We hypothesize that it implicitly regularizes the Lipschitz condition of the critic network especially over a local region where the model is uncertain (i.e., *a model-uncertain local region*). Therefore, the Bellman operator induced by the learned transition dynamics will yield an update to the agent's value function close to the true underlying Bellman operator's update. We provide systematic experimental results and theoretical analysis to support our hypothesis.

Based on this insight, we propose two simple yet effective robust training mechanisms to regularize the Lipschitz condition of the value network. The first one is *spectral normalization* (Miyato et al., 2018) which provides a global Lipschitz constraint for the value network. It directly follows our theoretical insights to explicitly control the Lipschitz constant of the value function. However, based on both of our theoretical and empirical observations, only the local Lipschitz constant around a model-uncertain local region is required for a good performance. So we also propose the second mechanism, *robust regularization*, which stabilizes the local Lipschitz condition of the value network by computing the adversarial noise with fast gradient sign method (FGSM) (Goodfellow et al., 2015). To compare the effectiveness of controlling global versus local Lipschitz, systematic experiments are implemented. Experimental results on five MuJoCo environments verify that the proposed Lipschitz regularization mechanisms with a single deterministic dynamics model improves SOTA performance of the probabilistic ensemble MBRL algorithms with less computational time.

Our contributions are summarized as follows. **(1)** We propose a new insight into why an ensemble of probabilistic dynamics models can significantly improve the performance of the MBRL algorithms, supported by both theoretical analysis and experimental evidence. **(2)** We introduce two robust training mechanisms for MBRL algorithms, which directly regularize the Lipschitz condition of the agent's value function. **(3)** Experimental results on five MuJoCo tasks demonstrate the effectiveness of our proposed mechanisms and validate our insights, improving SOTA asymptotic performance using less computational time and resources.

## 2   PRELIMINARIES AND BACKGROUND

**Reinforcement learning.** We consider a Markov Decision Process (MDP) defined by the tuple $(\mathcal{S}, \mathcal{A}, \mathcal{P}, \mathcal{P}_0, r, \gamma)$, where $\mathcal{S}$ and $\mathcal{A}$ are the state space and action space respectively, $\mathcal{P}(s'|s, a)$ is the transition dynamics, $\mathcal{P}_0$ is the initial state distribution, $r(s, a)$ is the reward function and $\gamma$ is the discount factor. In this paper, we focus on value-based reinforcement learning algorithms. Define the *optimal Bellman operator* $\mathcal{T}^*$ such that $\mathcal{T}^*Q(s, a) = r(s, a) + \gamma \int \mathcal{P}(s'|s, a) \arg\max_{a'} Q(s', a')ds'$. The goal of the value-based algorithm is to learn a $Q_\phi$ to approximate the optimal state-action value function $Q^*$, where $Q^* = \mathcal{T}^*Q^*$ is the fixed point of the optimal Bellman operator. For model-based RL (MBRL), we denote the approximate model for state transition and reward function as $\hat{\mathcal{P}}_\theta$ and $\hat{r}_\theta$ respectively. Similarly, we define the *model-induced Bellman Operator* $\widehat{\mathcal{T}}^*$ such that $\widehat{\mathcal{T}}^*Q(s, a) = \hat{r}(s, a) + \gamma \int \hat{\mathcal{P}}_\theta(s'|s, a) \arg\max_{a'} Q(s', a')ds'$.

**Probabilistic dynamics model ensemble.** Probabilistic dynamics model ensemble consists of $K$ neural networks $\hat{\mathcal{P}}_\theta = \{\hat{\mathcal{P}}_{\theta_1}, \hat{\mathcal{P}}_{\theta_2}, ..., \hat{\mathcal{P}}_{\theta_K}\}$ with the same architecture but randomly initialized with different weights. Given a state-action pair $(s, a)$, the prediction of each neural network is a Gaussian distribution with diagonal covariances of the next state: $\hat{\mathcal{P}}_{\theta_k}(s'|s, a) = \mathcal{N}(\mu_{\theta_k}(s, a), \Sigma_{\theta_k}(s, a))$. The model is trained using negative log likelihood loss (Janner et al., 2019): $\mathcal{L}(\theta_k) = \sum_{t=1}^{N} [\mu_{\theta_k}(s_t, a_t) - s_{t+1}]^\top \Sigma_{\theta_k}^{-1}(s_t, a_t)[\mu_{\theta_k}(s_t, a_t) - s_{t+1}] + \log \det \Sigma_{\theta_k}(s_t, a_t)$ During model rollouts, the probabilistic dynamics model ensemble first randomly selects a network from the ensemble and then samples the next state from the predicted Gaussian distribution.

**Local Lipschitz constant.** We now give the definition of local Lipschitz constant, a key concept in the following discussion.

**Definition 2.1.** *Define the **local** $(\mathcal{X}, \epsilon)$-**Lipschitz constant** of a scalar valued function $f : \mathbb{R}^N \to \mathbb{R}$ over a set $\mathcal{X}$ as:*

$$L^{(p)}(f, \mathcal{X}, \epsilon) = \sup_{x \in \mathcal{X}} \sup_{y_1, y_2 \in B^{(p)}(x, \epsilon)} \frac{|f(y_1) - f(y_2)|}{\|y_1 - y_2\|_p} \quad (y_1 \neq y_2). \tag{1}$$

*Throughout the paper, we consider $p = 2$ unless explicitly stated. If $L(f, \mathcal{X}, \epsilon) = C$ is finite, we say that $f$ is $(\mathcal{X}, \epsilon)$-locally Lipschitz with constant $C$.*

In particular, we can view the **global Lipschitz constant** as $L(f, \mathbb{R}^N, \epsilon)$, which upper bounds the local Lipschitz constant $L(f, \mathcal{X}, \epsilon)$ with $\mathcal{X} \subset \mathbb{R}^N$.

## 3 LOCAL LIPSCHITZ CONDITION OF VALUE FUNCTIONS IN MBRL

### 3.1 KEY INSIGHT: IMPLICIT REGULARIZATION ON VALUE NETWORK BY PROBABILISTIC ENSEMBLE MODEL

In this section, we provide insight into why a value-based reinforcement learning algorithm trained from simulated samples generated by a probabilistic ensemble model performs well in practice. We first conduct an experiment on the MuJoCo Humanoid environment with MBPO (Janner et al., 2019), the SOTA model-based algorithm using Soft Actor-Critic (SAC) (Haarnoja et al., 2018) as the backbone algorithm for policy and value optimization. We run MBPO with four different types of trained environment models: (1) a single deterministic dynamics model, (2) a single probabilistic dynamics model, (3) an ensemble of seven deterministic dynamics models, and (4) an ensemble of seven probabilistic dynamics models. As we can see from Figure 1a, in the Humanoid environment, the agent trained from an ensemble of the probabilistic dynamics model indeed achieves much better performance. A similar observation is found across all the other MuJoCo environments.

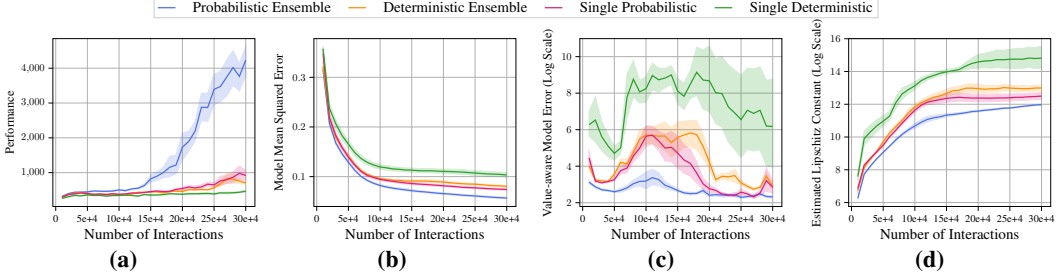

**Figure 1:** **(a)** Performance of MBPO algorithm trained with single deterministic, single probabilistic, deterministic ensemble, and probabilistic ensemble model respectively on Humanoid. **(b)** Model mean squared error. **(c)** Value-aware model error in log scale. **(d)** Upper bound of Lipschitz constant of value network. Results are averaged over 8 random seeds.

**Value-aware model error.** Why does MBPO with an ensemble of probabilistic dynamics models perform much better? We try to answer this question by considering the effects of the learned transition models on the agent's value functions. In MBPO algorithm, it fits an ensemble of $M$ Gaussian environment models $\hat{\mathcal{P}}_\theta^{(i)}(\cdot|s, a) \equiv \mathcal{N}\big(\mu_\theta^{(i)}(s, a), \Sigma_\theta^{(i)}(s, a)\big), i = 1, ..., M$. With a deep neural network as a powerful function approximator for the mean and variance of the Gaussian distribution, the mean squared error $\mathbb{E}_{(s,a)\sim\rho, s'\sim\mathcal{P}(\cdot|s,a), \hat{s}'\sim\hat{\mathcal{P}}_\theta(\cdot|s,a)}[\|s' - \hat{s}'\|^2]$ is often small. However, as argued in many previous MBRL works (Grimm et al., 2020; 2021), the mean squared error is often not a good measurement for the quality of the learned transition model. Instead, the *value-aware model error* (Farahmand et al., 2017a) as defined below plays a crucial role here, which connects directly to the suboptimality of the MBRL algorithm.

$$\mathcal{L}_{vame}(\hat{\mathcal{P}}_\theta; Q, \rho) = \mathbb{E}_{(s,a)\sim\rho}\big[\big(\mathcal{T}^*Q(s, a) - \widehat{\mathcal{T}}^*Q(s, a)\big)^2\big]$$

$$= \gamma^2 \mathbb{E}_{(s,a)\sim\rho}\Big[\Big(\int \mathcal{P}(s'|s, a)V(s')ds' - \int \hat{\mathcal{P}}_\theta(\hat{s}'|s, a)V(\hat{s}')d\hat{s}'\Big)^2\Big]$$

$$\leq \gamma^2 \mathbb{E}_{(s,a)\sim\rho, s'\sim\mathcal{P}(\cdot|s,a), \hat{s}'\sim\hat{\mathcal{P}}_\theta(\cdot|s,a)}\big[\big(V(\hat{s}') - V(s')\big)^2\big], \quad V(s) = \max_a Q(s, a)$$

This value-aware model error measures the difference between the simulated and true Bellman operator acting on a given $Q$ function. In other words, even though the learned transition model $\hat{\mathcal{P}}_\theta$ is approximately accurate, the model prediction error can still be amplified by the value function so that the two Bellman operators yield completely different updates on the value function.

**Value function Lipschitz regulates value-aware model error.** When the value function's local-Lipschitz constant is under control, the value-aware model error can be made small. As we see from Figure 1b, 1c, and 1d, although the probabilistic ensemble model and single deterministic model

achieve similar mean squared errors, value functions trained from an ensemble of probabilistic dynamics model has a significantly smaller Lipschitz constant and value-aware model error. (Note that the value-aware model error plotted in Figure 1d is in log scale.)

To see why this is the case, we view the learned Gaussian transition models $\hat{\mathcal{P}}_\theta^{(i)}$ as $f_\theta^{(i)} + g_\theta^{(i)}$, where $f_\theta^{(i)}$ is a deterministic model and $g_\theta^{(i)}(\cdot|s,a) \equiv \mathcal{N}(0, \Sigma_\theta^{(i)}(s,a))$ is a noise distribution with zero mean. When training the value network of the MBRL agent, the target value is computed as $r(s,a) + \gamma \mathbb{E}_{i \sim \text{Cat}(M, \frac{1}{M}), \epsilon_i \sim g_\theta^{(i)}(\cdot|s,a)}[V(f_\theta^{(i)}(s,a) + \epsilon_i)]$. Here we view data generated from the probabilistic ensemble model as an implicit augmentation. The augmentation comes from two sources: (i) variation of prediction across different models in the ensemble and (ii) the noise added by each noise distribution $g_\theta^{(i)}$. By augmenting the transition with a diverse set of noise and then training the value functions with such augmented samples, it implicitly regularizes the local Lipschitz condition of the value network over the local region around the state where the model prediction is uncertain.

In the next section, we provide some theoretical insights into how the local Lipschitz constant can play a role in the suboptimality of the MBRL algorithms. Later in the algorithm and experiment section, we will provide two practical mechanisms to regularize the Lipschitz condition of the value network and demonstrates the effectiveness of such mechanisms, which further validates our claim.

## 3.2 Error Bound of Model-based Value Iteration with locally Lipschitz value functions

In this section, we formally analyze how the Lipschitz constant of the value function affects the learning dynamics of the model-based (approximate) value iteration algorithm, the prototype for most of the value-based MBRL algorithms. For simplicity, following previous works (Farahmand et al., 2017b; Grimm et al., 2020), we assume that we have the true reward function $r(s,a)$, but extending the results to reward function approximation should be straightforward.

We consider the following value iteration algorithm. At $k$ th-iteration of the algorithm, we obtain a dataset $\mathcal{D}_k \triangleq \{(s_i, a_i, s_i')\}_{i=1}^N$, where $(s_i, a_i)$ is sampled i.i.d. from $\rho \in \Delta(\mathcal{S} \times \mathcal{A})$, the empirical state-action distribution, and $s_i' \sim \mathcal{P}(\cdot|s_i, a_i)$ is the next state under the environment transition. Based on this dataset, we first approximate the transition kernel $\hat{\mathcal{P}}$ to minimize the mean $L_2$ difference between the predicted and true next states.

$$\hat{\mathcal{P}}_k \leftarrow \underset{\hat{\mathcal{P}} \in \mathcal{M}}{\arg\min} \frac{1}{N} \sum_{i=1}^N \int \hat{\mathcal{P}}(\hat{s}'|s_i, a_i) \|\hat{s}' - s_i'\| d\hat{s}' \tag{2}$$

Then at $k$ th-iteration, we update the value function by solving the following regression problem:

$$\hat{Q}_k \leftarrow \underset{\hat{Q} \in \mathcal{F}}{\arg\min} \mathcal{L}_{reg}(\hat{Q}; \hat{Q}_{k-1}, \hat{\mathcal{P}}_k) := \underset{\hat{Q} \in \mathcal{F}}{\arg\min} \mathbb{E}_{(s,a) \sim \rho} \left[ \left( \hat{Q}(s,a) - \hat{\mathcal{T}}^* \hat{Q}_{k-1}(s,a) \right)^2 \right] \tag{3}$$

Empirically, we update the value function such that

$$\hat{Q}_k \leftarrow \underset{\hat{Q} \in \mathcal{F}}{\arg\min} \frac{1}{N} \sum_{i=1}^N \left| \hat{Q}(s_i, a_i) - \left( r_i + \gamma \int \hat{\mathcal{P}}_k(\hat{s}'|s_i, a_i) \hat{V}_{k-1}(\hat{s}') d\hat{s}' \right) \right|^2 \tag{4}$$

where $\hat{V}_{k-1}(\hat{s}') = \max_{a'} \hat{Q}_{k-1}(\hat{s}', a')$.

To simplify the analysis, we assume that we are given a fixed state-action distribution $\rho$ such that for every iteration, we can sample i.i.d. from this data distribution. However, in practice, we may use a different data collecting policy at different iterations. As argued in (Farahmand et al., 2017a), a similar result can be shown in this case by considering the mixing behavior of the stochastic process.

Now we list the assumptions we make. Some assumptions are made only to simplify the finite sample analysis, while others characterize the crucial aspects of model and value learning.

First, we make the deterministic assumption of the environment. This is only for the purpose of the finite sample analysis. When the environment transition is stochastic, we will not have the finite sample guarantee, but our insights still hold. We will provide more discussion on this later when we present our finite sample guarantee.

**Assumption 3.1.** *The environment transition is deterministic.*

Next, to apply the concentration inequality in our analysis, we have to make the following technical assumption of the state space and reward function.

**Assumption 3.2.** *(Boundedness of State Space and Reward Function) There exists constants $D$, $R_{\max}$ such that for all $s \in \mathcal{S}, a \in \mathcal{A}$, $\|s\|_2 \leq D$ and $r(s,a) < R_{\max}$.*

In addition, we make the following mild approximate realizability assumption on the model class of approximated transition kernels so that in the model space, at least one transition model close to the true underlying transition kernel should exist.

**Assumption 3.3.** $\big((\epsilon, \rho)$-*Approximate Realizability*$\big)$

$$\inf_{\hat{\mathcal{P}} \in \mathcal{M}} \mathcal{L}_2(\hat{\mathcal{P}}) := \inf_{\hat{\mathcal{P}} \in \mathcal{M}} \int \hat{\mathcal{P}}(d\hat{s}'|s,a) \big\| \mathcal{P}(s,a) - \hat{s}' \big\|_2 d\rho(s,a) \leq \epsilon \tag{5}$$

We also make a critical assumption of the local Lipschitz condition of the value function class. In particular, for the state-action value function $Q : \mathcal{S} \times \mathcal{A} \to \mathbb{R}$, we define that $Q$ is $(\epsilon, p)$-locally Lipschitz with constant $L$ if for every $a \in \mathcal{A}$, the function $Q_a : s \mapsto Q(s,a)$ is $(\epsilon, p)$-locally Lipschitz with constant less than or equal to L.

**Assumption 3.4.** *(Local Lipschitz condition of value functions) There exists a finite $L$, such that for every $Q \in \mathcal{F}$, it is $\big(\mathcal{X}, 2\epsilon\big)$-locally Lipschitz with a constant less than or equal to L, where $\mathcal{X}$ is the support of the distribution $\rho$.*

Note here we only need to assume that value functions are all $(\mathcal{X}, (1+\beta)\epsilon)$-locally Lipschitz with $\beta > 0$. We set $\beta = 1$ for simplicity. Finally, same as the previous work (Farahmand et al., 2017a), to provide a finite sample guarantee of model learning, we make the following assumption on the complexity of the model space of the approximated transition kernels

**Assumption 3.5.** *(Complexity of Model Space) Let $R > 0$, $J : \mathcal{M}_0 \to [0, \infty)$ be a pseudo-norm, where $\mathcal{M}_0$ is a space of transition kernels. Let $\mathcal{M} = \mathcal{M}_R = \{\mathcal{P} : J(\mathcal{P}) \leq R\}$. There exists constants $c > 0$ and $0 < \alpha < 1$ such that for any $\epsilon, R > 0$ and all sequence of state-action pairs $z_{1:n} \triangleq z_1, ..., z_n \in \mathcal{S} \times \mathcal{A}$, the following metric entropy condition is satisfied:*

$$\log \mathcal{N}(\epsilon, \mathcal{M}, L_2(z_{1:n})) \leq c \Big(\frac{R}{\epsilon}\Big)^{2\alpha} \tag{6}$$

*where $\mathcal{N}(\epsilon, \mathcal{M}, L_2(z_{1:n}))$ is the covering number of $\mathcal{M}$ with respect to the empirical norm $L_2(z_{1:n})$ such that $\|\mathcal{P}\|_{2,z_{1:n}}^2 = \dfrac{1}{N} \sum_{i=1}^{N} \|\mathcal{P}(\cdot|z_i)\|_2^2$*

Under these assumptions, we can now present our main theorem, which relates the local Lipschitz constant to the suboptimality of the approximate model-based value iteration algorithm. In particular, we provide a finite sample analysis of model learning in Theorem B.2 and value-aware model error in Theorem B.4. Then we apply the error accumulation results from (Farahmand et al., 2017a), connecting the local Lipschitz constant with the suboptimality of the algorithm.

**Theorem 3.6.** *Suppose $\hat{Q}_0$ is initialized such that $\hat{Q}_0(s,a) \leq \dfrac{R_{\max}}{1-\gamma}$ for $\forall\,(s,a)$. Under the assumptions of 3.1, 3.2, 3.3, 3.4, and 3.5, after $K$ iterations of the model-based approximate value iteration algorithm, there exists a constant $\kappa(\alpha)$ which depends solely on $\alpha \in (0,1)$ such that,*

$$\mathbb{E}_{(s,a)\sim\mathcal{P}_0}\Big[\big|Q^*(s,a) - \hat{Q}_K(s,a)\big|\Big] \leq \frac{2\gamma}{(1-\gamma)^2}\Big[C(\rho, \mathcal{P}_0)\Big(\max_{0 \leq k \leq K} \delta_k + \gamma^2\big(4\epsilon^2 L^2 \xi$$
$$+ \frac{(1-\xi)R_{\max}^2}{(1-\gamma)^2}\big)\Big) + 2\gamma^K R_{\max}\Big] \tag{7}$$

*where $\delta_k^2 = \mathcal{L}_{reg}(\hat{Q}_k; \hat{Q}_{k-1}, \hat{\mathcal{P}}_k)$ is the regression error defined in Equation (3), $\xi = 1 - \exp(-\dfrac{\epsilon N^{\frac{1}{1+\alpha}}}{\kappa(\alpha)D^2 R^{\frac{2\alpha}{1+\alpha}}})$, and $C(\rho, \mathcal{P}_0)$ is the concentrability constant defined in Definition A.1.*

**Remarks.**
**(1)** As the number of samples $N \to \infty$, $\xi \to 1$. Consequently, the value-aware model error $\mathbb{E}\big[|\mathcal{T}^* Q_k(s,a) - \widehat{\mathcal{T}}^* Q_k(s,a)|^2\big]$ will be governed by the term $4\epsilon^2 L^2 \xi$ which is controlled by the local Lipschitz constant $L$. In addition, if the number of iterations $k \to \infty$, the left hand side of

Equation (7) will be bounded by $\frac{2\gamma C(\rho, \mathcal{P}_0)}{(1-\gamma)^2}\left(\max_{0 \le k \le K} \delta_k + 4\epsilon^2 L^2 \xi\right)$. From here, we can see the role that the local Lipschitz constant has played in controlling the suboptimality of the algorithm. **(2)** Although we assume that the environment transition is deterministic, our insights should still hold when it is stochastic. In particular, if the learned transition model has a low prediction error $\mathcal{L}_2(\hat{\mathcal{P}})$ (Eq. 5) (which is often the case with a neural network as the function approximator), and the local Lipschitz constant of the value function is bounded, then we can still have a small value-aware model error and get a similar error propagation result as Theorem 3.6.

**Tradeoff between regression error and value-aware model error.** This theorem also reveals an essential trade-off between the regression error and value-aware model error through *the Lipschitz condition of the value function class*, i.e., the constant $L$. As $L$ gets smaller, the model-induced Bellman operator gets closer to the actual underlying Bellman operator. However, with a smaller $L$, we also impose a stronger condition on the value function class. Therefore, the value function space will shrink, and the regression error $\delta_k$ is expected to get larger.

To further visualize this trade-off, we conduct an experiment on the Inverted-Pendulum environment, where we run the analyzed model-based approximate value iteration algorithms for 1000 iterations. We plot (Figure 2a) the best evaluation performance for 20 episodes at every iteration, (Figure 2b) the maximum regression error across every iteration, as well as (Figure 2c) the maximum value-aware model error. To upper bound the (local) Lipschitz constant, we constrain the spectral norm of the weight matrix for each layer of the value network.

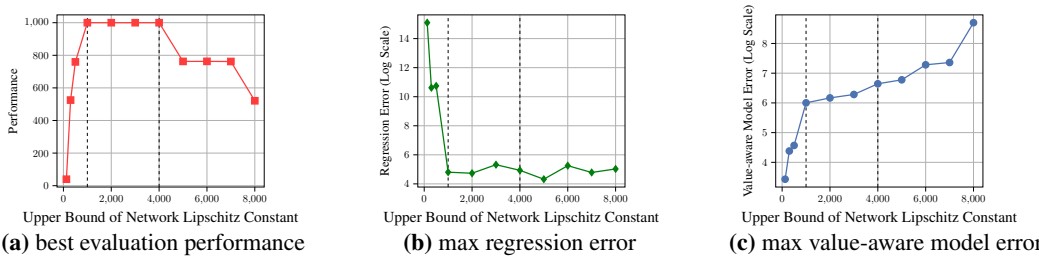

**(a)** best evaluation performance     **(b)** max regression error     **(c)** max value-aware model error

**Figure 2:** Model-based value iteration on Inverted-Pendulum across value networks with different Lipschitz constraints. All the results are the median over 5 random seeds.

As we see from the results in Figure 2b, indeed, the regression error dramatically drops as the Lipschitz constant grows from 100 to 1000 and then levels off, which indicates that perhaps the Lipschitz constant of 1000 is rich enough for the value function class to be Bellman-complete for this environment. Meanwhile, the value-aware model error also increases with a bigger Lipschitz constant. However, we can achieve a good balance between these two errors when the Lipschitz constants are between 1000 (left dashed line) and 4000 (right dashed line), where the algorithm is observed to perform the best.

## 4 METHODS

In this section, we present two different approaches to regularize the (local) Lipschitz constant of the value function.

### 4.1 SPECTRAL NORMALIZATION

First, to explicitly control the upper bound of the Lipschitz constant, we adapt the technique of Spectral Normalization, which is originally proposed to stabilize the training of Generative Adversarial Network (GAN) (Miyato et al., 2018). By controlling the spectral norm of the weight matrix at every layer of the value network, its Lipschitz constant is upper bounded. In particular, during each forward pass, we approximate the spectral norm of the weight matrix $\|W\|_2$ with one step of power iteration. Then, we perform a projection step so that its spectral norm will be clipped to $\beta$ if bigger than $\beta$, and unchanged otherwise. See Appendix C for more computational details.

### 4.2 ROBUST REGULARIZATION

Spectral Normalization directly bounds the global Lipschitz constant of the value function. However, as we argued in Section 3.2, we only require the local Lipschitz conditions around the model-

uncertain local region. Such a strong regularization is not necessary and can even negatively impact the expressive power of value function. Now we present an alternative approach to quickly regularize the local Lipschitz constant of the value function based on a robust training mechanism.

To regularize the local Lipschitz condition of the value network, we minimize the following loss:

$$\mathcal{L}_{\text{reg}}^{(\alpha)}(Q_\phi; \epsilon, \pi_\psi, \{s_i, a_i, s_i'\}_{i=1}^D) = \sum_{i=1}^D \max_{\|\tilde{s}_i - s_i\|_\alpha \leq \epsilon} \left( Q_\phi\big(\tilde{s}_i, \pi_\psi(s_i')\big) - Q_\phi(s_i, a_i) \right)^2 \tag{8}$$

This robust loss is to guarantee that the variation of the value function locally is small. Then we combine it with the original loss of the value function $\mathcal{L}_{org}(Q_\phi; \pi_\psi, \{s_i, a_i, s_i'\}_{i=1}^D)$ by minimizing $\mathcal{L}(Q_\phi; \pi_\psi, \{s_i, a_i, s_i'\}_{i=1}^D) = \mathcal{L}_{org}(Q_\phi; \pi_\psi, \{s_i, a_i, s_i'\}_{i=1}^D) + \lambda \mathcal{L}_{\text{reg}}^{(\alpha)}(Q_\phi; \epsilon, \pi_\psi, \{s_i, a_i, s_i'\}_{i=1}^D).$

Here, $\lambda$ and $\epsilon$ are two hyperparameters. A larger $\epsilon$ is required for the dynamics model which has larger prediction errors. In terms of $\lambda$, a bigger $\lambda$ makes the value network varies smoother over the local region and gives better convergence guarantees, but it also hurts the expressive power of the value network. In practice, we find that using $\epsilon = 0.1$ is often enough for good performance. So we fix $\epsilon$ and search for the best $\lambda$. For a detailed discussion on the choice of $\lambda$ s, see Appendix D.4.

To solve the perturbation $\tilde{s}$ in the constrained optimization within the robust loss, we use the fast gradient sign method (FGSM) (Goodfellow et al., 2015). Here, let $(s, a)$ be the state action pair. Then we compute $\tilde{s} = \text{proj}\left(s_0 + \epsilon \, \text{sign}\left(\nabla_{\tilde{s}}\big|_{\tilde{s}=s_0}\big(Q_\phi\big(\tilde{s}, \pi_\psi(\tilde{s})\big) - Q_\phi(s, a)\big)^2\right), B^{(\alpha)}(s_0, \epsilon)\right),$ where $s_0 \sim \mathcal{N}(s, \epsilon^2 I)$. Here the purpose of random initialization of $s_0$ is because the gradient $\nabla_{s'}\big(Q_\phi\big(\tilde{s}, \pi_\psi(\tilde{s})\big) - Q_\phi(s, a)\big)^2)$ vanishes at $s$. In our experiments, we focus on the $l_\infty$ norm, but our proposed robust loss should be applicable to any $l_p$ norm.

In addition, we can apply more advanced constrained optimization solvers such as Projected Gradient Decent (PGD). But here, using FGSM together with a single deterministic environment model is mainly for efficiency purposes. In practice, we find that using PGD does not give much performance gain over FGSM method, and we refer the readers to Appendix D.3 for the experimental results.

## 5 EXPERIMENT

### 5.1 EMPIRICAL EVALUATIONS OF PROPOSED MECHANISMS

We evaluate our two proposed training mechanisms on five MuJoCo tasks, including Walker, Humanoid, Ant, Hopper, and HalfCheetah. We compare the two training mechanisms with MBPO (Janner et al., 2019) using an ensemble of probabilistic transition models. We do not compare our methods with MBPO using an ensemble of deterministic transition models mainly because it is ensemble-based and across all five tasks, it is outperformed by probabilistic ensemble models. In addition, since our regularization mechanisms are only trained on top of a single deterministic transition model, we compare it with both MBPO using a single deterministic transition model and a single probabilistic model. We implement our methods and the baseline methods based on a PyTorch implementation of MBPO (Lin, 2022). More implementation details are provided in Appendix C.

**Improved asymptotic performance.** Figure 3 presents the learning curves for all methods. These results show that using just a single deterministic model, MBPO with our two Lipschitz regularization mechanisms achieves a comparable and even better performance across all five tasks than MBPO with a probabilistic ensemble model. In particular, the proposed robust regularization technique shows a larger advantage on three more sophisticated tasks: Humanoid, Ant, and Walker. For example, on Humanoid, it achieves the same final performance as that of a probabilistic ensemble with only about 60% of the environment interaction steps. In contrast, spectral normalization shows little improvement over a single deterministic model on these three tasks, showing the limitation of constraining the global Lipschitz constant. We provide more discussion on this in Section 5.3.

**Time Efficiency.** Compared with MBPO using an ensemble of probabilistic models, our proposed training mechanisms, especially robust regularization, is more time efficient. In Table 1 we record the wall clock time of the algorithms trained for 200,000 environment interaction steps on Walker and Ant, averaged over 8 random seeds. We see that robust regularization is much faster than spectral normalization. While spectral normalization takes slightly less the amount of time than probabilistic ensemble, robust regularization only takes about 70% of the computational time.

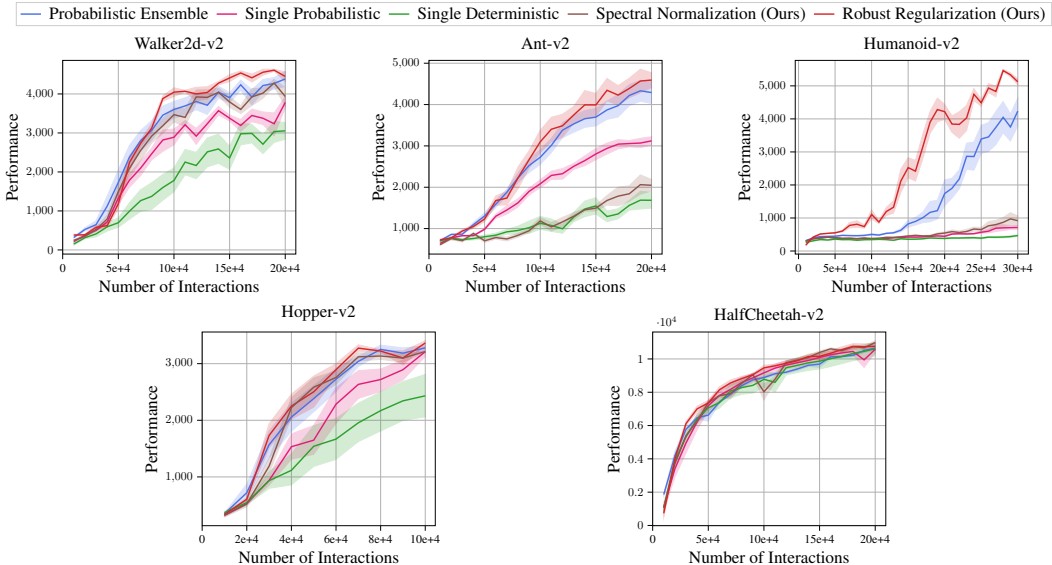

**Figure 3:** Performance of the proposed value function training mechanisms against baselines. Results are averaged over 8 random seeds and shaded regions correspond to the 95% confidence interval among seeds.

The significance of the experimental results is twofold. First, it further validates our insights and shows the importance of the local Lipschitz condition of value functions in MBRL. Second, it demonstrates that having an ensemble of transition models is not necessary. We can save the computational time and cost of training an ensemble of transition models with a simple Lipschitz regularization mechanism. In practice, we can even use a single probabilistic model combined with our proposed mechanisms to get the best performance. See Appendix D.2 for the extra experimental results.

**Table 1:** Comparison of computational time.

| Algorithm | Walker | Ant |
|---|---|---|
| | Time (h) | Time(h) |
| Single Deterministic | 29.6 | 34.0 |
| Probabilistic Ensemble | 60.0 | 62.3 |
| **Robust Regularization** | 45.4 | 48.3 |
| **Spectral Normalization** | 54.2 | 57.1 |

## 5.2 VISUALIZING THE VALUE-AWARE MODEL ERROR

Given the excellent performance of robust regularization, we now verify its effectiveness in controlling the value-aware model error. On Walker, we compare it with a variant: computing the perturbation with uniform random noise instead of adversarially choosing the perturbation.

As we can see from the first figure in Figure 4, robust regularization has a much smaller value-aware model error than all other methods. Adding uniformly random noise can somewhat reduce the value-aware model error compared with a single deterministic model (without any noise added). However, it is still much less effective than robust regularization, which computes the error adversarially. In the second figure of Figure 4, we see that with uniform random noise, it achieves a slightly better performance than a single deterministic model but is still far worse than robust regularization. The results further verify that by adversarially choosing the noise, robust regularization is extremely effective at controlling the value-aware model error, resulting in great empirical performance. In Appendix D.1, we provide additional experimental results of value-aware model error in the rest of the four environments.

## 5.3 ROBUST REGULARIZATION VS. SPECTRAL NORMALIZATION

In Table 1, we already see that robust regularization achieves a much better time efficiency than spectral normalization. Now we analyze their performance difference through the lens of value-aware model error. To reduce the value-aware model error, we only need to control the local Lipschitz constant of the value network over the model-uncertain region, and thus controlling the global Lipschitz constant of the value function with spectral normalization is not necessary. To see this, in the last two figures of Figure 4 , we visualize the value-aware model error and performance of the algorithm on Walker with varying spectral radius $\beta$, defined in Section 4.1.

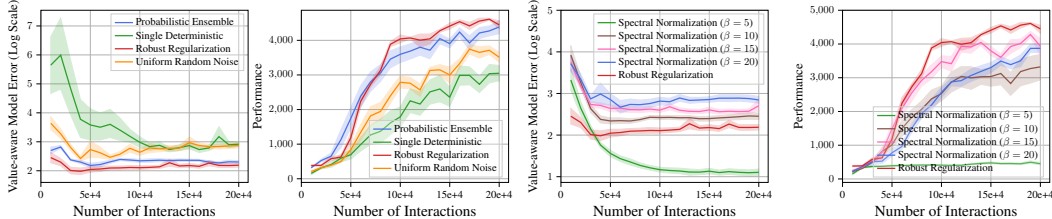

**Figure 4:** (Left Two) Robust Regularization with a comparison of uniform random noise on Walker. (Right Two) Spectral Normalization with different spectral radius $\beta$ on Walker.

As we see from the plot, to reduce the value-aware model error so that it is about the same as robust regularization, we need to choose the spectral radius $\beta$ as small as 10. However, under this constraint, it achieves a significantly worse performance than robust regularization, indicating that this constraint is perhaps too strong. To get the best performance, spectral normalization has to use a larger spectral radius to trade value-aware model error for expressive power and thus has worse empirical performance. In addition, we observe from Figure 3 that the performance discrepancy between the two methods is even greater in more complicated environments such as Ant and Humanoid, which further shows the limitation of constraining the global Lipschitz constant.

## 6 RELATED WORK

**Dynamics ensemble in model-based reinforcement learning.** Dynamics model ensemble is first introduced in MBRL by Kurutach et al. (2018) to avoid the learned policy exploit the insufficient data regions. Then Chua et al. (2018) proposed a probabilistic dynamics ensemble to capture the aleatoric uncertainty and epistemic uncertainty, and achieves significant performance improvement compared to the deterministic dynamics ensemble. Probabilistic dynamics ensemble is now widely used in MBRL methods (e.g. Lai et al. (2020); Clavera et al. (2020); D'Oro & Jaśkowski (2020); Lai et al. (2021); Froehlich et al. (2022); Li et al. (2022)). However, despite its popularity, there is still no clear explanation why the use of probabilistic dynamics ensemble brings such a large improvement to the policy. In this paper, we fill this gap and propose a novel theoretical explanation on why probabilistic dynamics ensemble works well.

**Lipschitz continuity in reinforcement learning.** In the realm of model-free RL, a few recent works also apply the technique of spectral normalization to regularize the Lipschitz condition of the RL agent's value function, which results in more stable optimization dynamics (Gogianu et al., 2021; Bjorck et al., 2021). Ball & Roberts (2021) investigate the effects of added Gaussian noise of exploration on the smoothness of value functions. In addition, Shen et al. (2020); Zhang et al. (2020) propose to use a similar robust regularizer as our robust regularization method, aiming at an adversarially robust policy. In our work, we focus on MBRL, a fundamentally different setting. As explained in Section 3.1 and Appendix F, if no restrictions are imposed on the value function class, then the value-aware model error could explode, whereas the model error vanishes in the model-free setting. In MBRL, Osband & Van Roy (2014) connects Lipschitz constant of the value function to the regret bound of posterior sampling for RL. Asadi et al. (2018) proposed to learn a generalized Lipschitz transition model with respect to the Wasserstein metric, resulting in a bounded multi-step model prediction error and Lipschitz optimal value function induced from the learned dynamics. In contrast, our paper focuses on the local Lipschitz condition of the value function instead of the underlying transition model, studying its relation with the suboptimality of MBRL algorithm.

## 7 CONCLUSION AND DISCUSSION

In this paper, we provide insight into why the probabilistic ensemble model can achieve great empirical performance. We demonstrate the importance of the local Lipschitz condition of value functions in MBRL algorithms and justify our hypothesis with both theoretical and empirical results. Based on our insight, we propose two training mechanisms that directly regularize the Lipschitz condition of the value function. Empirical studies demonstrate the effectiveness of the proposed mechanisms. One limitation is that if we use a single model instead of an ensemble, then we cannot use many existing methods of uncertainty estimation based on model ensemble, and we need to redesign the uncertainty estimation mechanism based on a single model. We leave this as future work.

ACKNOWLEDGEMENT

This work is supported by National Science Foundation NSF-IIS-FAI program, DOD-ONR-Office of Naval Research, DOD Air Force Office of Scientific Research, DOD-DARPA-Defense Advanced Research Projects Agency Guaranteeing AI Robustness against Deception (GARD), Adobe, Capital One and JP Morgan faculty fellowships.

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

# Supplementary Material

## A  ADDITIONAL DEFINITIONS AND NOTATION

We start with the definition of the Concentrability constant

**Definition A.1.**  *(**Concentrability Constant**) (Farahmand et al., 2017a)*
*Given $\rho, \nu \in \Delta(\mathcal{S})$, an integer $k > 0$, and an arbitrary sequence of policies $(\pi_i)_{i=1}^k$, the distribution $\rho \mathcal{P}^{\pi_1} \mathcal{P}^{\pi_2} ... \mathcal{P}^{\pi_k}$ denotes the future state distribution obtained when the state in the first step is distributed according to $\rho$ and the agent follows the sequence of policies $\pi_1, ..., \pi_k$. Define:*

$$c_{\rho,\nu}(k) = \sup_{\pi_1,...,\pi_k} \left\| \frac{d\rho \mathcal{P}^{\pi_1} \mathcal{P}^{\pi_2} ... \mathcal{P}^{\pi_k}}{d\nu} \right\|_{2,\nu}$$

*Here, $\|f\|_{2,\nu}^2 = \int f(s)^2 d\nu$. The derivative $\dfrac{d\rho \mathcal{P}^{\pi_1} \mathcal{P}^{\pi_2} ... \mathcal{P}^{\pi_k}}{d\nu}$ is the Radon-Nikydom Derivative of two probability measures, which is well-defined up to a set of measure zero by $\nu$ if $\rho \mathcal{P}^{\pi_1} \mathcal{P}^{\pi_2} ... \mathcal{P}^{\pi_k}$ is absolutely continuous with respect to $\nu$. In case it's not absolutely continuous, we set it to be $\infty$. Then, for a constant $0 \le \gamma < 1$, define the discounted weighted average concentrability coefficient as*

$$C(\rho, \nu) = (1 - \gamma)^2 \sum_{k=1}^{\infty} \gamma^{k-1} c_{\rho,\nu}(k)$$

Throughout the proof, we use $\|f\|_{p,\mu}$ denote the $L_p(\mu)$-norm $1 \le p < \infty$ of a measurable function $f : \mathcal{S} \to \mathbb{R}$ such that

$$\|f\|_{p,\mu}^p = \int_{\mathcal{S}} |f(x)|^p d\mu(x)$$

In addition, we define the empirical norm. Given a collection points $\{s_1, ..., s_n\}$ in $\mathcal{S}$, define the empirical norm $L_p(s_1, ..., s_n)$ such that

$$\|f\|_{L_p(s_1,...,s_n)}^p = \frac{1}{N} \sum_{i=1}^{N} |f(s_i)|^p$$

Finally, we define Rademacher complexity as in (Bartlett et al., 2005). Same as Farahmand et al. (2017a), we will use a local variant of Rademacher complexity to derive the rate of estimation error.

**Definition A.2.** *(Rademacher Complexity) Let $\sigma_1, ..., \sigma_n$ be $n$ independent Rademacher random variables, i.e. $\mathbb{P}\{\sigma_i = -1\} = \mathbb{P}\{\sigma_i = 1\} = \frac{1}{2}$. Given a collection of measurable functions $\mathcal{F}$ from $\mathcal{X}$ to $\mathbb{R}$ and a probability distribution $\mu$ over $\mathcal{X}$, we sample n points $x_1, ..., x_n$ i.i.d. from $\mu$. Define*

$$R_n \mathcal{F} = \sup_{f \in \mathcal{F}} \frac{1}{N} \sum_{i=1}^{n} \sigma_i f(x_i)$$

*Then we define the Rademacher complexity of $\mathcal{F}$ as $\mathbb{E}[R_n \mathcal{F}]$*

Besides, same as (Bartlett et al., 2005), we define the sub-root function as non-negative and non-decreasing function $\psi : [0; \infty) \to [0, \infty)$ such that $r \mapsto \frac{\psi(r)}{\sqrt{r}}$ is non-increasing for $r > 0$

## B  PROOF OF THE THEOREM

We begin by citing the following theorem.

**Theorem B.1.** *(Bartlett et al., 2005) Let $\mathcal{F}$ be a class of functions with values in range $[a, b]$ and assume that there are some functional $T : \mathcal{F} \to \mathbb{R}^+$ and some constant $B$ such that for every $f \in \mathcal{F}$,*

$$Var[f] \le T(f) \le B\mathbb{E}(f) \tag{9}$$

*Let $\psi$ be a sub-root function and let $r^* = r^*(\mathcal{F})$ be the fixed point of $\psi$. Assume that for any $r \le r^*$, $\psi$ satisfies*

$$\psi(r) > B\mathbb{E}[R_n \{f \in \mathbb{F} : T(f) \le r\}] \tag{10}$$

*Then, with $c_1 = 704$ and $c_2 = 26$, for any $K > 1$ and every $x > 0$, with probability at least $1 - e^{-x}$, for any $f \in \mathcal{F}$, we have*

$$\mathbb{E}[f] \leq \frac{K}{K-1}\mathbb{E}_n[f] + \frac{c_1 K}{B}r^* + \frac{x(11(b-a) + c_2 BK)}{n} \tag{11}$$

*Also with a probability at least $1 - e^{-x}$, for any $f \in \mathcal{F}$, we have*

$$\mathbb{E}_n[f] \leq \frac{K}{K-1}\mathbb{E}[f] + \frac{c_1 K}{B}r^* + \frac{x(11(b-a) + c_2 BK)}{n} \tag{12}$$

*We will then use $r^\star(\mathcal{F})$ to denote the fixed point of a sub-root function $\psi$ that satisfies 10*

Now we prove the following theorem which provides a finite sample bound on the value-aware model error.

**Theorem B.2.** *Under the four assumptions 3.1, 3.2, 3.3, 3.5, with the probability model learned based on Equation 2, there exists a constant $\kappa(\alpha)$ which depends solely on $\alpha \in (0, 1)$, such that with probability $1 - \delta$,*

$$L(\hat{\mathcal{P}}) \leq \epsilon + \frac{\kappa(\alpha)D^2 R^{\frac{2\alpha}{1+\alpha}}\ln(\frac{1}{\delta})}{N^{\frac{1}{1+\alpha}}}, \tag{13}$$

*where $N$ is the number of samples from the data-collection distribution $\rho$, $D$ is the size of the state-space defined in assumption 3.2, and $R$ is defined in the metric entropy condition of the model class in assumption 3.5.*

*Proof.* Given a batch of state-action transition triples $\{(s_i, a_i, s_i')\}_{i=1}^N$ with $(s_i, a_i)$ sampled i.i.d from data distribution $\rho$, we denote the empirical loss

$$L_n(\mathcal{P}) = \frac{1}{N}\sum_{i=1}^N \int_{\mathcal{S}} \mathcal{P}(\hat{s}_i'|s_i, a_i)\|\hat{s}_i' - s_i'\|d\hat{s}_i' \tag{14}$$

We also denote the underlying loss over the data distribution $\rho$ as

$$L(\mathcal{P}) = \mathbb{E}_{(s,a)\sim\rho}\Big[\int_{\mathcal{S}} \mathcal{P}(\hat{s}'|s, a)\|\hat{s}' - s'\|d\hat{s}'\Big] \tag{15}$$

In addition, let $\tilde{\mathcal{P}} \in \mathcal{M}$ be the best model in the transition kernel class $\mathcal{M}$ defined in 3.5.

Now we would like to apply Theorem B.2 to bound the difference between the empirical and the true underlying loss. First, let $\mathcal{F} = \{(s \times a, s') \mapsto l(s \times a, s'; \mathcal{P}) - l(s \times a, s'; \tilde{\mathcal{P}}); \mathcal{P} \in \mathcal{M}\}$ be the class of functions in Theorem B.1, where $l(s \times a, s'; \mathcal{P})$ is the single datapoint version of the empirical loss $L_n(\mathcal{P})$. So $\mathbb{E}_{s\times a\sim\rho}[l(s \times a, s'; \mathcal{P})] = L(\mathcal{P})$. Now by assumption 3.2, $0 \leq l(s \times a, a; \mathcal{P}) \leq 2D$ for every $s \times a \in \mathcal{S} \times \mathcal{A}, s' \in \mathcal{S}$, and $\mathcal{P} \in \mathcal{M}$. Therefore, the value of $f$ is bounded between $-2D$ and $2D$ for every $f \in \mathcal{F}$,

As a consequence, for every $f \in \mathcal{F}$, $Var(f) \leq \mathbb{E}[f^2] \leq 4D^2$. So we can set

$$T(f) = 4D^2 \mathbb{E}\Big[\int_{\mathcal{S}} \mathcal{P}(\hat{s}'|s, a)\|\hat{s}' - s'\|d\hat{s}'\Big]$$
$$B = 4D^2$$

Now, we can apply Theorem B.2 to conclude that with probability $1 - \delta$ (let $K = 2$),

$$L(\mathcal{P}) - L(\tilde{\mathcal{P}}) \leq 2(L_n(\mathcal{P}) - L_n(\tilde{\mathcal{P}})) + \frac{2 \times 704}{4D^2}r^*(\mathcal{F}) + \frac{(11 \times 4D + 2 \times 26 \times 4D^2)\ln(\frac{1}{\delta})}{N} \tag{16}$$

Since $\hat{\mathcal{P}}$ is the minimizer of the empirical loss $L_n(\mathcal{P})$,

$$L(\hat{\mathcal{P}}) - L(\tilde{\mathcal{P}}) \leq \frac{352}{D^2}r^*(\mathcal{F}) + \frac{(44D + 208D^2)\ln(\frac{1}{\delta})}{N} \tag{17}$$

We can provide an upper bound of the local Rademacher complexity $r^*(\mathcal{F})$: there exists a finite constant $\tau > 0$ such that for a given $0 \leq \alpha \leq 1$, we have

$$r^*(\mathcal{F}) \leq \frac{c_1(\alpha)D^4 R^{\frac{2\alpha}{1+\alpha}}}{N^{\frac{1}{1+\alpha}}} + \frac{\tau D^4 \ln N}{N}, \tag{18}$$

where $c(\alpha) = \dfrac{\tau}{(1-\alpha)^{\frac{2}{1+\alpha}}}$. The proof follows the exact same steps of Proposition 10 in Farahmand et al. (2017a). Now back to Equation 17, by the realizability assumption 3.3, the best model $\tilde{\mathcal{P}}$ in the model class satisfies that $L(\tilde{\mathcal{P}}) \leq \epsilon$. Therefore, with probability $1 - \delta$,

$$L(\hat{\mathcal{P}}) \leq \epsilon + \frac{352 c_1(\alpha)D^2 R^{\frac{2\alpha}{1+\alpha}}}{N^{\frac{1}{1+\alpha}}} + \frac{352 \tau D^2 \ln N}{N} + \frac{(44D + 208D^2)\ln(\frac{1}{\delta})}{N} \tag{19}$$

Finally, there should exist a constant $\kappa(\alpha)$ sufficiently large such that with probability $1 - \delta$,

$$L(\hat{\mathcal{P}}) \leq \epsilon + \frac{\kappa(\alpha)D^2 R^{\frac{2\alpha}{1+\alpha}} \ln(\frac{1}{\delta})}{N^{\frac{1}{1+\alpha}}} \tag{20}$$

**Corollary B.3.** *Under the five assumptions 3.1, 3.2, 3.3, and 3.5 with the probability model learned based on Equation 2, there exists a constant $\kappa(\alpha)$ which depends solely on $\alpha \in (0,1)$, such that with probability $1 - \exp(-\dfrac{\epsilon N^{\frac{1}{1+\alpha}}}{\kappa(\alpha)D^2 R^{\frac{2\alpha}{1+\alpha}}})$,*

$$L(\hat{\mathcal{P}}) \leq 2\epsilon, \tag{21}$$

*Proof.* This is a straightforward application of Theorem B.2, where we could just let $\epsilon = \dfrac{\kappa(\alpha)D^2 R^{\frac{2\alpha}{1+\alpha}} \ln(\frac{1}{\delta})}{N^{\frac{1}{1+\alpha}}}$.

Next, we consider the local Lipschitz condition of the value function and provide a finite sample bound of the value-aware model error.

**Theorem B.4.** *Under the five assumptions 3.1, 3.2, 3.3, 3.4, and 3.5, with the probability model learned based on Equation 2, there exists a constant $\kappa(\alpha)$ which depends solely on $\alpha \in (0,1)$, such that for any $m > 1$,*

$$\int \left| \mathcal{T}^* Q(s,a) - \widehat{\mathcal{T}}^* Q(s,a) \right|^2 d\rho(s,a) \leq \gamma^2 \Big[ 4\epsilon^2 L^2 \xi + \frac{R_{\max}^2}{(1-\gamma)^2}(1-\xi) \Big], \tag{22}$$

*where $\xi = 1 - \exp(-\dfrac{\epsilon N^{\frac{1}{1+\alpha}}}{\kappa(\alpha)D^2 R^{\frac{2\alpha}{1+\alpha}}})$*

*Proof.*

$$\begin{aligned}
\|\mathcal{T}^* Q - \widehat{\mathcal{T}}^* Q\|_\rho^2 &= \int \left| r(s,a) + \gamma V(s') - r(s,a) - \gamma \int \hat{\mathcal{P}}(d\hat{s}'|s,a)V(\hat{s}') \right|^2 d\rho(s,a) \\
&= \gamma^2 \int \left| \int \hat{\mathcal{P}}(d\hat{s}'|s,a)\big(V(\hat{s}') - V(s')\big) \right|^2 d\rho(s,a) \\
&\leq \gamma^2 \int \int \mathcal{P}(ds'|s,a)\big(V(\hat{s}') - V(s')\big)^2 d\rho(s,a) \\
&\leq \gamma^2 \int \int \mathbb{1}\{\|s' - \hat{s}'\| \leq 2\epsilon\}\big(V(\hat{s}') - V(s')\big)^2 \hat{\mathcal{P}}(d\hat{s}'|s,a)d\rho(s,a) \\
&\quad + \gamma^2 \int \int \mathbb{1}\{\|s' - \hat{s}'\| > 2\epsilon\}\big(V(\hat{s}') - V(s')\big)^2 \hat{\mathcal{P}}(d\hat{s}'|s,a)d\rho(s,a) \\
&\leq \gamma^2 2^2 \epsilon^2 L^2 \mathbb{P}\{\|s' - \hat{s}'\| \leq 2\epsilon\} + \gamma^2 \frac{R_{\max}^2}{(1-\gamma)^2}\mathbb{P}\{\|s' - \hat{s}'\| > 2\epsilon\}
\end{aligned}$$

Now apply Corollary B,

$$\|\mathcal{T}^*Q - \widehat{\mathcal{T}}^*Q\|_\rho^2 \leq \gamma^2 4\epsilon^2 L^2 \big(1 - \exp(-\frac{\epsilon N^{\frac{1}{1+\alpha}}}{\kappa(\alpha)D^2 R^{\frac{2\alpha}{1+\alpha}}})\big)$$
$$+ \gamma^2 \frac{R_{\max}^2}{(1-\gamma)^2} \exp(-\frac{\epsilon N^{\frac{1}{1+\alpha}}}{\kappa(\alpha)D^2 R^{\frac{2\alpha}{1+\alpha}}})$$

Finally, with the value-aware model error bounded, we could apply the error propagation results from (Munos, 2005; Farahmand et al., 2017a) and prove our main theorem, which relates the local Lipschitz constant to the sub-optimality of the approximate model-based value iteration algorithm.

**Theorem B.5.** *Suppose $\hat{Q}_0$ is initialized such that $\hat{Q}_0(s,a) \leq \dfrac{R_{\max}}{1-\gamma}$ for $\forall\,(s,a)$. Under the assumptions of 3.1, 3.2, 3.3, 3.4, and 3.5, after $K$ iterations of the model-based approximate value iteration algorithm, there exists a constant $\kappa(\alpha)$ which depends solely on $\alpha \in (0,1)$ such that,*

$$\mathbb{E}_{(s,a)\sim\mathcal{P}_0}\Big[\big|Q^*(s,a) - \hat{Q}_K(s,a)\big|\Big] \leq \frac{2\gamma}{(1-\gamma)^2}\Big[C(\rho,\mathcal{P}_0)\Big(\max_{0\leq k\leq K}\delta_k + \gamma^2\big(4\epsilon^2 L^2\xi$$
$$+ \frac{(1-\xi)R_{\max}^2}{(1-\gamma)^2}\big)\Big) + 2\gamma^K R_{\max}\Big] \quad (23)$$

*where $\delta_k^2 = \mathcal{L}_{reg}(\hat{Q}_k; \hat{Q}_{k-1}, \hat{\mathcal{P}}_k)$ is the regression error defined in Equation (3), $\xi = 1 - \exp(-\dfrac{\epsilon N^{\frac{1}{1+\alpha}}}{\kappa(\alpha)D^2 R^{\frac{2\alpha}{1+\alpha}}})$, and $C(\rho,\mathcal{P}_0)$ is the concentrability constant defined in Definition A.1.*

*Proof.* This follows directly from the Theorem B.4 and also Theorem 4 from (Farahmand et al., 2017a), where the value-aware model error $e_{model}(N) \leq \gamma^2\Big(4\epsilon^2 L^2\xi + \dfrac{R_{\max}^2}{(1-\gamma)^2}(1-\xi)\Big)$

## C IMPLEMENTATION DETAILS

In this section, we are going to introduce the implementation details of our two proposed methods. We implement our methods based on a PyTorch implementation of MBPO (Lin, 2022). The dynamics model architecture is MLP with four hidden layers of size 200. In Ant and Humanoid, the hidden size is 400 because these two environments are more complex than others. For the probabilistic dynamics model ensemble, we set the ensemble size to 7 which is the setting used in the original paper of MBPO (Janner et al., 2019). The policy is optimized with Soft Actor-Critic (SAC) (Haarnoja et al., 2018). The actor network architecture and the critic network architecture are MLP with two hidden layers of size 256.

For robust regularization, we fix $\epsilon$ to 0.1 as discussed in Section 4.2, and we do a grid search of $\lambda$ over $[0.01, 0.1, 1.0]$, with the result presented in Table 2 For spectral normalization, we add the normalization on each layer of the critic network. In particular, at every forward pass, we approximate the spectral radius of the weight matrix with one step of power iteration. The algorithm is sketched below with $\boldsymbol{u}$ and $\boldsymbol{v}$ being the right and left singular vector of the weight matrix $W$.

$$\boldsymbol{v} \leftarrow W\boldsymbol{u}^{(t-1)}; \quad \alpha \leftarrow \|\boldsymbol{v}\|; \quad \boldsymbol{v}^{(t)} \leftarrow \alpha^{-1}\boldsymbol{v} \quad (24)$$
$$\boldsymbol{u} \leftarrow W^T\boldsymbol{v}^{(t)}; \quad \rho \leftarrow \|\boldsymbol{v}\|; \quad \boldsymbol{u}^{(t)} \leftarrow \rho^{-1}\boldsymbol{u} \quad (25)$$

Then we perform a projection of the parameters: $W := \max(1, \dfrac{\max(\alpha,\rho)}{\beta})^{-1}W$. So the spectral norm will be clipped to $\beta$ if it's bigger than $\lambda$, unchanged otherwise. We do a grid search of $\beta$ over $[15, 20, 25, 30, 35]$, with results shown in Table 3. For a fair comparison, we set the rollout horizon during model rollouts to 1 in all environments.

**Table 2:** Robust Regularization with different regularization weights $\lambda$

| Environment | Probabilistic Model Ensemble | Deterministic Ensemble | $\lambda = 0.01$ | $\lambda = 0.01$ | $\lambda = 1.0$ |
|---|---|---|---|---|---|
| Walker | 3236 | 4385 | 4138 | **4515** | 4112 |
| HalfCheetah | 10519 | 10648 | 10871 | **10982** | 10763 |
| Ant | 1684 | 4290 | 4190 | **4592** | 4011 |
| Humanoid | 468 | 4235 | 4210 | 4531 | **5123** |
| Hopper | 2432 | 3278 | **3213** | 3110 | 3010 |

**Table 3:** Spectral Normalization with different spectral radius $\beta$

| Environment | $\beta = 15$ | $\beta = 20$ | $\beta = 25$ | $\beta = 30$ | $\beta = 35$ |
|---|---|---|---|---|---|
| Walker | 3870 | **4447** | 4124 | 3982 | 3645 |
| HalfCheetah | 9931 | 10452 | **10763** | 10313 | 10423 |
| Ant | 3870 | **4447** | 4124 | 3982 | 3645 |
| Humanoid | 427 | 662 | 704 | **712** | 684 |
| Hopper | 2876 | 3125 | **3364** | 3243 | 2972 |

# D    ADDITIONAL EXPERIMENT RESULTS

## D.1    ADDITIONAL EXPERIMENTAL RESULTS OF VALUE-AWARE MODEL ERROR

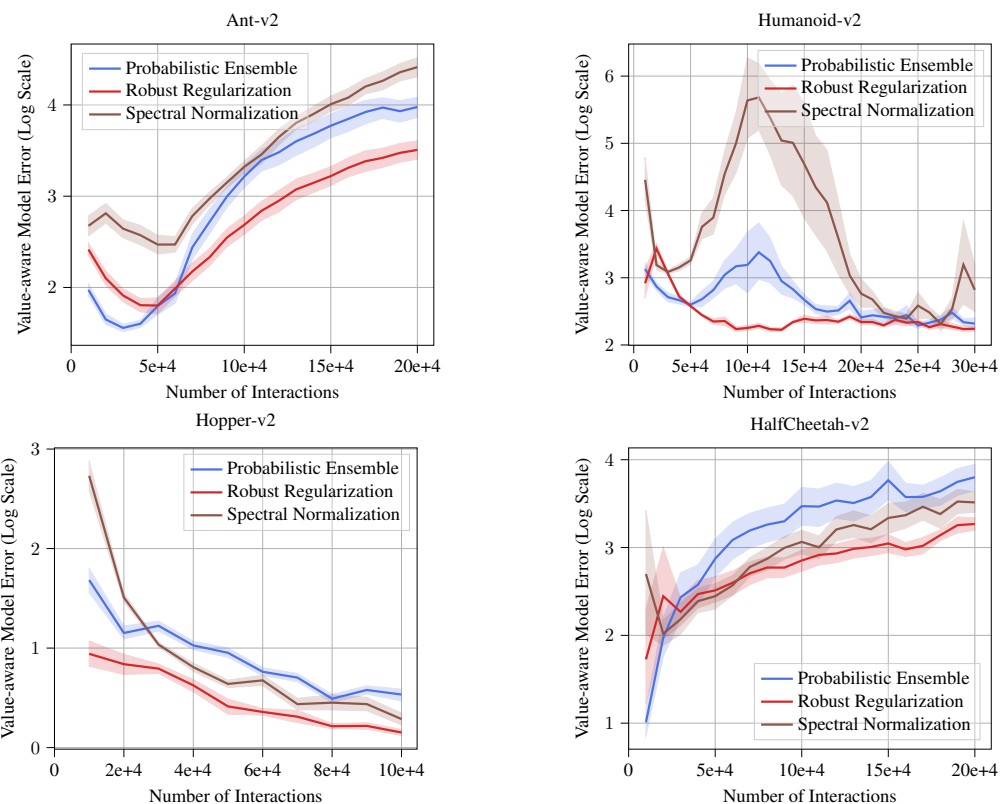

**Figure 5:** Comparison between the proposed mechanisms and the probabilistic ensemble baseline in terms of Value-aware Model Error (Log Scale)

In Section 5.2 and 5.3, we demonstrate the effectiveness of robust regularization in controlling the value-aware model error on Walker and also show the limitation of constraining the global Lipschitz constant by spectral normalization. Here, we provide the results of value-aware model error in the rest of the four environments. We used the same best hyperparameters reported in Table 2 and Table 3, which have the best performance under grid search. Once again, we observe that robust

regularization effectively reduces the value-aware model error by constraining the local Lipschitz condition with computing adversarial perturbation. In addition, we also see that global Lipschitz constraints are too strong for spectral normalization. In two more complicated environments, Ant and Humanoid, it has to sacrifice value-aware model error for the expressive power of the value function. Therefore, spectral normalization does not achieve a good performance in these two environments. However, in two easier environments, Hopper and HalfCheetah, spectral normalization could still effectively reduce the value-aware model error and has good empirical performance.

## D.2 PROPOSED MECHANISMS WITH SINGLE PROBABILISTIC MODEL

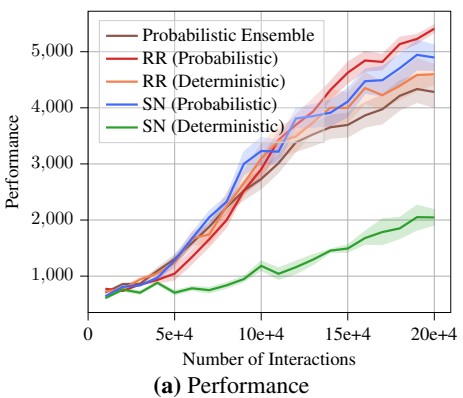
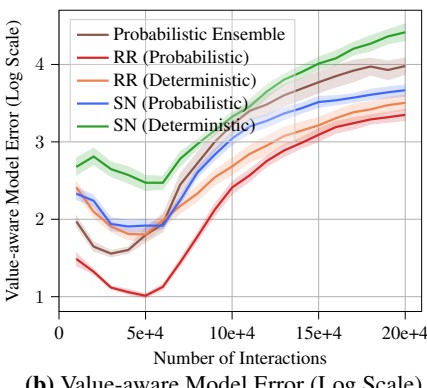

**(a)** Performance        **(b)** Value-aware Model Error (Log Scale)

**Figure 6:** Spectral Normalization and Robust Regularization with a single probabilistic model on Ant. RR is short for Robust Regularization, and SN is short for Spectral Normalization

In the experiment section, we combine our proposed mechanisms with a single deterministic model and compare it against MBPO using an ensemble of probabilistic models. The purpose is to verify that regularization of the local Lipschitz constant is critical in MBRL algorithms and propose a computationally efficient MBRL algorithm without a model ensemble. In practice, complementary to our proposed Lipschitz regularization mechanisms, we can also use a single probabilistic model to further regularize the local Lipschitz condition of the value function. In addition, training a probabilistic environment model would be better suited for environments with stochastic transitions.

Here in Figure 6, we combine our two proposed mechanisms with both a probabilistic and deterministic model on Ant, comparing them with the probabilistic ensemble baseline. From Figure 6b, we see that although spectral normalization with a single deterministic model has a large value-aware model error, it is significantly reduced when combining it with a probabilistic dynamics model. Therefore, we find that spectral normalization with a probabilistic model achieves much better performance and even outperforms MBPO with an ensemble of probabilistic models. For robust regularization, using a probabilistic model also helps improve the algorithm's value-aware model error and performance. This observation suggests that the two Lipschitz regularization approaches, explicit regularization by spectral normalization or robust regularization and implicit regularization by probabilistic models, are complementary. In practice, we can combine the two approaches to get the best performance of the MBRL algorithm.

## D.3 ROBUST REGULARIZATION WITH FGSM VS. PGD

In Figure 7, we compare the performance of robust regularization with 20 steps of Project Gradient Descent (PGD-20) against the Fast Gradient Sign Method (FGSM) on Walker. In particular, although PGD-20 is much more computationally expensive, we do not observe the improvement with this more powerful constrained optimization solver.

## D.4 ROBUST REGULARIZATION WITH DIFFERENT REGULARIZATION WEIGHTS

Figure 8 further visualizes how the regularization weight $\lambda$ of robust regularization influences the algorithm performance. Similar to the findings of the experiments on spectral normalization, we see that the algorithm's performance first increases and drops as the regularization weight gets larger. This verifies our theoretical insights from Theorem B.5 that with a small $\lambda$, with algorithm gets less

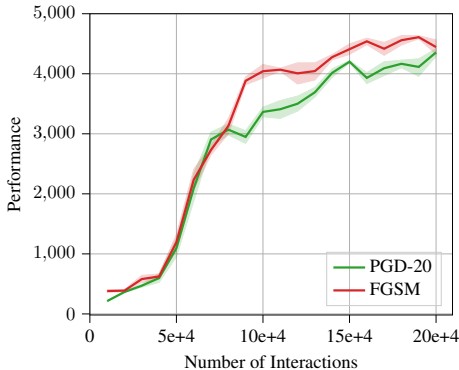

**Figure 7:** Robust Regularization with 20 steps of Project Gradient Descent (PGD-20) against Fast Gradient Sign Method (FGSM).

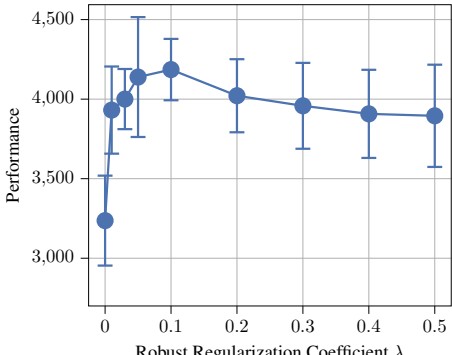

**Figure 8:** Robust Regularization with different regularization weights $\lambda$'s on Walker. Experiments are all with 8 random seeds.

regularization and thus has a big value-aware model error. But meanwhile, it also has a small regression error since the regularization has little effect on the expressive power of the value function. When $\lambda$ goes up, the regularization will have a stronger negative effect on the expressive power of the value function, but the value-aware model error will also get smaller. We observe that the algorithm performs the best with $\lambda = 0.1$, achieving the balance between value-aware model error and the value function's expressive power.

# E  ADDITIONAL DETAILS ON THE INVERTED-PENDULUM EXPERIMENT

In Section 3.2, we provide an experiment of model-based value iteration on the Inverted Pendulum to further verify the validity of our theorem. Below we provide the pseudocode for it.

---

**Algorithm 1** Model-based Approximate Value Iteration on Inverted Pendulum

---

$K$ : Total number of iterations for the algorithm
$H$ : Number of gradient steps to solve the inner optimization
$\mathcal{M}, \mathcal{G}$ : Space of transition probability kernels and reward functions
$\mathcal{F}$: Space of value function
$\mu \in \Delta(\mathcal{S} \times \mathcal{A})$: a data-collecting state-action distribution
Sample i.i.d from $\mu$ to generate the training dataset $\mathcal{D} = \{(s_i, a_i, s_i', r_i)\}_{i=1}^N$

$$\hat{\mathcal{P}} \leftarrow \underset{\hat{\mathcal{P}} \in \mathcal{M}}{\arg\min} \sum_{i=1}^N \|s_i' - \int \hat{\mathcal{P}}(ds'|s,a)s'\|^2$$

$$\hat{r} \leftarrow \underset{\hat{r} \in \mathcal{G}}{\arg\min} \sum_{i=1}^N (r_i - \hat{r}(s_i, a_i))^2$$

Initialize the value function $\hat{Q}_0$.
**repeat**
  **for** $k = 0$ **to** $K - 1$ **do**
    Sample i.i.d $N$ state-action pairs from $\rho : \{(s_i, a_i)\}_{i=1}^N$
    Compute $\hat{s}_i' \sim \mathcal{P}(\cdot|s_i, a_i), \hat{r}_i = \hat{r}(s_i, a_i)$
    **for** $t = 0$ **to** $H - 1$ **do**

      Update policy using gradient ascent with $\dfrac{1}{N} \sum_{i=1}^N \nabla_\theta \hat{Q}_\phi(s_i, \pi_\theta(s_i))$

    **end for**
    **for** $t = 0$ **to** $H - 1$ **do**
      Update value function using gradient descent with
      $$\frac{1}{N} \sum_{i=1}^N \nabla_\phi \left( \hat{r}_i + \gamma \hat{Q}_\phi(\hat{s}_i', \pi_\theta(\hat{s}_i')) - \hat{Q}_\phi(s_i, a_i) \right)^2$$
    **end for**
  **end for**
**until** end of training
**Output:** $\hat{Q}_\phi$

---

# F  LIPSCHITZ REGULARIZATION OF MODEL-FREE RL ALGORITHMS

**Figure 9:** Combination of robust regularization with model-free SAC

In this paper, we provide both theoretical and empirical insights into why Lipschitz regularization is crucial in model-based RL algorithms through the lens of value-aware model error. On the contrary, the model error vanishes in the model-free setting, so our theoretical insights no longer hold. However, fitting a value function with a smaller Lipschitz constant may still be beneficial for policy optimization and the value prediction of out-of-distribution state-action pairs.

So does the improvement shown in Section 5 indeed come from the controlled value-aware model error, which is unique in the model-based setting? We conduct an experiment on the model-free setting, adding our proposed spectral normallization and robust regularization mechanisms to the model-free SAC respectively with the same hyperparameter settings used in the paper. As shown in Figure 9, combining the Lipschitz regularization mechanisms slightly improves the performance of model-free SAC algorithm. Still, the improvement is far more limited compared with the improvement of the model-based algorithm shown in Figure 3. The results suggest that although some additional aspects of value learning could be affected by Lipschitz regularization, our insights into the value-aware model error for model-based scenarios should still hold. The exploration of how Lipschitz regularization impacts other aspects of value learning in the model-free setting is beyond the scope of our work, but it would be an interesting direction for future work.

