# OpenReview forum: "Is Model Ensemble Necessary? Model-based RL via a Single Model with Lipschitz Regularized Value Function"
_ICLR.cc/2023/Conference — ICLR 2023 poster_

### Official Review · Reviewer_24kG · 2022-10-14

**Confidence:** 3
**Correctness:** 3
**Technical Novelty And Significance:** 3
**Empirical Novelty And Significance:** 3
**Recommendation:** 8

**Clarity, Quality, Novelty And Reproducibility:**

**Clarity** -- The paper and results are clearly explained and presented.

**Quality** -- The mathematical results and experiments appear rigorous. The one suggestion here is to clarify if/when the theoretical results apply to stochastic MDPs.

**Novelty** -- The paper is novel to the best of my knowledge.

**Reproducibility** -- Most experiment details are contained in the appendix; some important details (e.g., the hyperparameters for FGSM) are omitted. Code is not included.

**Strength And Weaknesses:**

Strengths
* I think this question studied in this paper is really important. Both in model-based RL and in other subareas of RL, ensembles are used to great effect, but _why_ they're useful has remained an open problem.
* The empirical results are quite strong. It's great that the paper not only identifies the underlying mechanism that explains why ensembles are useful, but also uses that finding to propose a simpler approach.

Weaknesses
* Writing. The writing could be improved/clarified in some places (see below). The paper has a number of grammar errors. I'd recommend copy-pasting the paper into a Google Doc and running the grammar checker.
* It seems like the proposed method requires careful hyperparameter tuning, with different hyperparameters used for each environment. While some prior work does this, it is generally frowned upon. I would recommend highlighting this limitation in the main paper.


Questions/concerns
* Is this really a paper about model-based RL, or is it showing that smooth Q-functions are useful, especially when doing many gradient steps on each example? I.e., if this same regularizer is applied to a model-free RL method, can it match the performance of model-based methods?
* Stochastic dynamics (top of page 6) -- Do the results hold for stochastic dynamics? The discussion at the top of page 6 seems to indicate that they do, but the results on page 5 say that they do not.
* Eq. 9 -- Would a 1-sided penalty work, too? I.e., is the main purpose of the regularizer to make the Q-function smooth, or to prevent it from making significantly _larger_ predictions for nearby states?
* Why isn't the proposed method faster? I would have guessed that removing the ensemble would make the method 2 - 4x faster.

Minor writing comments
* "to validate" -> "to test" -- "validate" seems less rigorous
* Perhaps cite this paper [1], which provides a nice discussion of how noise in the actor can provide smoothness in the critic (w.r.t. the actions, rather than the states).
* The passive voice is to be avoided where possible.
* "than using only a single dynamics model" -- Cite.
* "we hypothesize" -> "We hypothesize"
* "Therefore, the Bellman operator ..." -- I didn't understand this sentence.
* "Ian J Goodfellow" -- This is an odd citation format.
* "Plug-and-play modules" -- Where is this shown in the experiments?
* "time and resources" -- what type of resources?
* "value function Q*" -- add a comma at the end
* "we define the model-induced Bellman Operator" -- Should this use $\hat{r}$ rather than $r$?
* Eq 1 -- This could be cut or moved to the appendix.
* "Define local" --> "Define the local"
* Eq. 2 -- What is the first "sup" over? Is that a typo?
* Sec 3.1, first paragraph -- the line spacing here looks odd.
* All figures -- Make the xlabel and ylabel the same size as the surrounding text. Use the Matplotlib built-in log-scale and scientific notation, rather than doing it manually (it is easier to read).
* In other words, even ..." -- This sentence is great!
* "achieve similar mean squared errors" -- This is fairly subjective; the ensemble does achieve lower errors. Perhaps just put numbers to it: "The mean squared errors are within x%"
* Assumption 3.1 seems very strict. I'd recommend adding more discussion of why this is needed and when this can be relaxed.
* Assumption 3.2 -- This is a very weak assumption, and perhaps could simply be noted when defining the MDP (and not stated as another assumption in this section)
* Eq 6 -- Is the norm here the L2 norm? If so, indicate that using "$\|\|_2$"
* "it's" -- It's generally better to avoid contractions in technical writing.
* "Assumption 3.5, $J(\mathcal{P})$ -- Where is $J(\cdots)$ defined?
* "Tradeoff between" -- This section is really great!
* "We plot (1) ..." -- It might be clearer to replace "(1)" with "(Fig 2a)"
* "between 1000 and 4000" -- Where does the 4000 number come from?
* "more advanced constrained optimization solvers such as PGD" -- To me, PGD seems no more complex than FGSM
* Fig 3 -- I found the color for "single probabilistic" very hard to read. Consider using colorblind-friendly colors, or adding different markers to the lines
* "then many existing methods ..." -- I didn't understand this sentence.

[1] https://arxiv.org/pdf/2101.11331.pdf

**Summary Of The Paper:**

This paper studies why ensembles of dynamics models are useful for model-based RL. The main result is that ensembles effectively regularize the value function to be more smooth, and explicitly regularizing the value function can achieve similar performance without requiring an (expensive) ensemble.

**Summary Of The Review:**

Overall, this is a strong paper that makes progress on an important problem. I think the paper could be improved in a few ways, but I nevertheless think that the paper should be accepted.

---

### Official Review · Reviewer_4iuU · 2022-10-23

**Confidence:** 4
**Correctness:** 3
**Technical Novelty And Significance:** 2
**Empirical Novelty And Significance:** 2
**Recommendation:** 6

**Clarity, Quality, Novelty And Reproducibility:**

As mentioned above, the setting and contribution of the paper are not clearly illustrated. The writing of the paper still requires improvement to make readers understand the results.

Some minor issues:
1. What is cat(K, 1/K) in section 3.1?
2. Main theorem should not involve any definition from the appendix

**Strength And Weaknesses:**

Strength: The paper aims to provide some insights for designing more efficient RL algorithms.
Weakness: The contribution of the paper is rather vague to readers. It is not very clear what the exact problem the paper wants to solve and what can we learn from the results. In particular, the concept of model ensemble in the title is not fully explained in the paper, and the later main results are based on very strong conditions, e.g., deterministic transition. In more details, I have the following questions:

1. What is 'model ensemble' in this paper exactly referring to? Is it an ensemble of multiple independent fitting or sequential adaptive fitting? Should the algorithm stated in section 3.2 be considered ensemble algorithm or a single model algorithm as mentioned in the title? One thing make me confused is that the number of ensemble models and number of value iterations are both $K$. Is this a correct?

2. Does the upper bound in Theorem 3.6 converges to zero as $K\rightarrow\infty$? If it does (under any condition), please clearly show it. If it does not, I can only see it decreases as the Lipschitz constant and $C$ getting smaller, which does not provide any new insight.

3. What is exactly hidden when we only consider deterministic transition rather than stochastic transition? Please be more clear about whether the analysis framework will go through and what will be different.



**Summary Of The Paper:**

The paper studies 'probabilistic dynamics model ensemble' method in reinforcement learning. It proposes that (1) an important factor to the convergence of RL is the Lipschitz condition of the value function and (2) model ensemble helps to regularize the Lipschitz condition in the training. The paper provides both theoretical analysis and experiment results to support such claims.

**Summary Of The Review:**

The paper is not ready for publication yet as the contribution of the paper is rather vague to readers.

---

> ### Comment · Area_Chair_zjgH · 2022-11-24
> **Thank you! Are you satisfied by the answers?**
>
> Dear reviewer,
>
> Thanks again for your detailed review! The authors have replied back to you. Please read them carefully, and acknowledge their response. If there is still an unclear point about the paper or you do not agree with some of the responses, please let them know. We would like to have a robust discussion now.
>
> If you have any further questions from them, please ask them now. We have to make the final decision soon.
> Also as a courtesy to the authors, please acknowledge their rebuttal.
>
> Thank you,
> Area Chair

---

### Official Review · Reviewer_S2Gs · 2022-10-24

**Confidence:** 4
**Clarity, Quality, Novelty And Reproducibility:** The work is clearly written, relevant…
**Correctness:** 4
**Technical Novelty And Significance:** 3
**Empirical Novelty And Significance:** 3
**Recommendation:** 8

**Strength And Weaknesses:**

Detailed review:

== Strengths ==

- The paper combines theoretical insight and empirical evidence in a very convincing fashion.
- The insight seems broadly useful for many related model based algorithms that use a Dyna style update rule.
- The practical algorithm seems to be a strong and simple addition to MBRL algorithms
- The paper is easy to understand, the mathematical results are well explained both with rigorous derivation and intuitive explanations.

== Weaknesses ==

Main main concern is the presentation and contextualization of the result. It seems that the impact of the Lipschitz regularity is independent of the concrete choice of an ensemble model. The theoretical derivation does not hinge on the fact that the model is represented by an ensemble and while the empirical results do show that the performance of the regularized algorithm is superior even when using a single model, this does not necessarily suggest that ensembles cannot contribute separately. For example the model can be used to construct certainty estimates or to drive exploration, which the underlying MBPO algorithm does not leverage.

I would strongly suggest that the authors decouple their core contribution, the impact of Lipschitz bounds of the value functions, from the insights into ensemble models in the presentation. This is not because I think that any of the claims are wrong (although I think the statement that "model ensembles might not be necessary" is slightly to general to be supported by the paper), but because I think their insight is more valuable as an independent insight, not just as an auxiliary insight to ensemble performance.

---

Empirically, I would like to see a comparison of the impact of the Lipschitz regularization on a model free baseline. This simple ablation would highlight the fact that it is the interplay of Lipschitz smoothness and model based learning that produces the effect, not an independent mechanism that improves value function learning in general.

In the related work section, the authors acknowledge previous work in the area of Lipschitz regularization in RL, but claim that they work in the "fundamentally different" field of MBRL. While I agree that the authors insights are important and novel in MBRL, I do not think that the field is "fundamentally different" enough from MFRL that the alternate hypothesis that spectral normalization simply improves RL (MF or MB) can be dismissed out of hand.

---

The regularization/spectral norm creates an important hyperparameter, which the authors themselves highlight as a tradeoff between value function flexibility and model optimization. I would have liked to see more discussion or experimental evaluation what happens with a sub-optimal choice for this hyperparameter. If the algorithm is not robust to this choice, it creates yet another potential pitfall in a field which already struggles with complicated algorithm which are very expensive to tune. The results in the appendix showcase at least some robustness to the concrete choice of regularization, but it is unclear whether this holds over all benchmark tasks. Since the grid-search was performed already, would it be possible to append the results?

---


== Minor corrections ==
- Page 6, top: For In particular -> In particular?
- Page 18, bottom: computationally expansive -> expensive?

**Summary Of The Paper:**

The paper "Is model ensemble necessary" presents an investigation of the interplay between value function smoothness (in the form of a Lipschitz constant) and model errors in model based reinforcement learning. They present a theoretical analysis and show empirical performance gains based on their insights in a practical algorithm.

**Summary Of The Review:**

The authors present a well written paper that carefully examines the impact of a Lipschitz regularization on the performance of model based reinforcement learning. They highlight both theoretically and empirically how the Lipschitz bound of the value functions impacts the performance of model based value estimation and how the insight can be transformed into a strong algorithm. Overall, I think the paper is a strong submission to ICRL. However, I believe there are issues in the presentation that could be improved to strengthen the submission further, with the details outlined above.

---

### Official Review · Reviewer_pnXv · 2022-10-25

**Confidence:** 2
**Clarity, Quality, Novelty And Reproducibility:** Paper was clear, experiments were nov…
**Correctness:** 3
**Technical Novelty And Significance:** 3
**Empirical Novelty And Significance:** 3
**Recommendation:** 6

**Strength And Weaknesses:**

I think the experimental results are quite good, and I really like that the authors provide a good path to get sample efficient results without an ensemble. I also think that value function regularization is a good path to follow.

The main weakness is the theoretical contributions.
- For a continuous function over a convex state space, the local $(\epsilon,p)$-Lipschitz constant is the same as the Lipschitz constant (see proof below). It would seem to be easier to simply assume the state space to be convex, and use a global Lipschitz constant. So why all the complexity with the Local-Lipschitz property. Or am I missing something here?
- I don't feel that Theorem 3.6 is particularly interesting, I would hope for something where the error goes down as N increases, but that's not the case. I fail to see why a bound that doesn't decrease as more environment interaction iterations occur is somehow useful, since it will never get better than the performance after just one iteration.

I think a good theoretical avenue would be to look at [1], which lays out a relationship between model error and RL performance, and also shows that a bounded Lipshcitz is important.

Proof that $L(\epsilon,p)(f)=L(\infty,p)(f)$. The authors already state that $\epsilon_1\leq\epsilon_2\Rightarrow L(\epsilon_1,p)(f) \leq L(\epsilon_2, p)(f)$, so I will show the $\epsilon_1\geq\epsilon_2\Rightarrow L(\epsilon_1,p)(f) \leq L(\epsilon_2, p)(f)$.

This follows simply from a property of continuous functions over a convex state space, if there are states $s_1,s_2$ with
$f(s_1)<f(s_2)$ and $s':=\lambda s_1 + (1-\lambda) s_2$ is some point between $s_1$ and $s_2$, then

$$\left\vert\frac{f(s_1)-f(s_2)}{\Vert s_1-s_2\Vert}\right\vert\leq\max\left(\left\vert\frac{f(s_1)-f(s')}{\Vert s_1-s'\Vert}\right\vert,\left\vert\frac{f(s')-f(s_2)}{\Vert s'-s_2\Vert}\right\vert\right)$$

so over a smaller distance (either from $s_1$ to $s'$ or from $s'$ to $s_2$) we get a larger lipschitz constant, showing my claim


[1] Ian Osband, Model-based Reinforcement Learning and the Eluder Dimension

**Summary Of The Paper:**

In this paper, the authors propose to regularize the value function approximation in MBRL in order to learn policies without having to use an ensemble of models (as is done in MBPO bu Janner).

After the authors present evidence that the ensemble of models is indeed regularizing the value function by using the value-aware model error, they propose their own scheme to regularize the value function.

This scheme works quite well in experimental results, beating MBPO while only using one model, not an ensemble of models.

**Summary Of The Review:**

This paper provides empirical evidence for a strong new avenue of research in MBRL: value function regularization. Although I find the theory unconvincing, I think the message of this paper is something the RL community would benefit from, providing empirical evidence that will invite theories to follow.

---

> ### Comment · Area_Chair_zjgH · 2022-11-24
> **Thank you! Are you satisfied by the answers?**
>
> Dear reviewer,
>
> Thanks again for your detailed review! The authors have replied back to you. Please read them carefully, and acknowledge their response. If there is still an unclear point about the paper or you do not agree with some of the responses, please let them know. We would like to have a robust discussion now.
>
> If you have any further questions from them, please ask them now. We have to make the final decision soon.
> Also as a courtesy to the authors, please acknowledge their rebuttal.
>
> Thank you,
> Area Chair

---

### Decision · Program_Chairs · 2023-01-20

**Decision:**

Accept: poster

**Justification For Why Not Higher Score:**

(1) This is a good paper, but perhaps not groundbreaking or very novel.
(2) The theoretical analysis misses some important discussions about the deterministic dynamics.
That being said, I wouldn't mind bumping up to a spotlight.

**Justification For Why Not Lower Score:**

All reviewers are positive about this paper.

**Metareview: Summary, Strengths And Weaknesses:**

**SUMMARY:**
The paper first empirically studies the role of ensemble of models in model-based RL and shows that they have a smaller value-aware model error. The result suggests that this is due to implicit regularization of the ensemble, leading a smaller Lipschitz constant of the estimated value function.
The paper then theoretically studies the effect of Lipschitz constant of the value function on the value-aware model error, suggesting a tradeoff involving the Lipschitz constant. Smaller Lipschitz constant of the value function implies a smaller effect of the model error, as indicated in the value-aware model error term of the upper bound in Theorem 3.6, but it can possibly causes a larger regression error.

Based on this insight, the paper suggests that one should control the Lipschitz constant of the value function. Two methods are suggested, both applicable to a single model, instead of an ensemble. One of the methods minimizes the spectral norm of the value network weights. This controls the global Lipschitz constant of the network. The other controls the local Lipschitz constant of the value function.

The paper performs several experiments showing that the proposed methods, especially the one controlling the local Lipschitz constant, leads to a performance comparable or superior to varieties of methods, including those using an ensemble of probabilistic models.

**EVALUATION:**
The reviewers are all positive about this work with scores ranging from 6 to 8. They believe that the paper brings new insight, the experimental results are good, and the suggested methods are simple additions to the arsenal of MBRL algorithms.

On the negative side, they mention that the presentation and contextualization can be improved. They also mention that the assumption of deterministic transition dynamics is strong.

I have read the paper, and I generally agree with the reviewers. I have two further comments.

The first comment is that as far as I see, the only evidence that an ensemble leads to a smaller Lipschitz constant is the empirical result of Figure 1(d). The argument of "By augmenting the transition with a diverse set of noise and ..." in the penultimate paragraph of Section 3.1 is not a formal proof. Is it possible to make this a theoretical result?

The second comment is related to the theory and in particular the deterministic transition assumption.
First of all, I should clarify that the concentrability constant should be $C(P_0, \rho)$ and not $C(\rho, P_0)$. Definition A.1 defines $C(\rho, \nu)$, in which $\rho$ is the initial state distribution and $\nu$ is the data distribution. In the main body of the paper, $P_0$ is the initial state distribution and $\rho$ is the data distribution. This is a minor issue.

The more important issue is that each term $c_{\rho,\nu}(k)$ in Definition A.1, which would be $c_{P_0, \rho}(k)$ according to the notation of the main body, has terms in the form of

$\frac{\mathrm{d} P_0 P^{\pi_1} ... P^{\pi_k} }{\mathrm{d} \rho}$,

the Radon-Nikodym derivative of these distributions. In order for this to be defined, the distribution in the numerator should be absolutely continuous with respect to the distribution rho. If we assume that rho has a density over the state space, it means that $P_0 P^{\pi_1} \cdots P^{\pi_k}$ should also have a density. When is this true?

If $P_0$ is a discrete measure (for instance, the initial state is always $s_0$), the distribution $P_0 P^{\pi_1} \cdots P^{\pi_k}$ is always discrete because of the deterministic dynamics. This means that the constant $c_{P_0, \rho}(k)$ would be infinity, making the upper bound meaningless. Some remarks and clarifications are required here.

Another minor issue is that there are two references Farahmand et al. 2017a and 2017b, both of them referring to the same paper. Some of those references should be to Farahmand, "Iterative Value-Aware Model Learning," NeurIPS, 2018 as several of the theoretical results are based on that paper, and not the 2017 one.

Overall,  this is a good paper and should be accepted.

**Note From Pc:**

if the above contains the word "oral" or "spotlight" please see: "oral" presentation means -> notable-top-5% and "spotlight" means -> notable-top-25%. As stated in our emails, we are disassociating presentation type from AC recommendations